# Ion Release and Surface Changes of Nickel–Titanium Archwires Induced by Changes in the pH Value of the Saliva—Significance for Human Health Risk Assessment

**DOI:** 10.3390/ma15061994

**Published:** 2022-03-08

**Authors:** Zana Jusufi Osmani, Borut Poljšak, Saša Zelenika, Ervin Kamenar, Kristina Marković, Marko Perčić, Višnja Katić

**Affiliations:** 1Faculty of Dental Medicine, University of Rijeka, 51000 Rijeka, Croatia; zanajusufiosmani@yahoo.com; 2Laboratory of Oxidative Stress Research, Faculty of Health Sciences, University of Ljubljana, 1000 Ljubljana, Slovenia; poljsakb@zf.uni-lj.si; 3Faculty of Engineering & Centre for Micro- and Nanosciences and Technologies, University of Rijeka, 51000 Rijeka, Croatia; sasa.zelenika@riteh.hr (S.Z.); ekamenar@riteh.hr (E.K.); kristina@riteh.hr (K.M.); mpercic@riteh.hr (M.P.)

**Keywords:** nickel hypersensitivity, orthodontic wires, surface coating, surface roughness

## Abstract

The aim of this study was to explore whether changes in the salivary pH influence mechanical properties, surface roughness, and ion release from NiTi archwires with various surface coatings, and discuss the clinical significance of the findings. The uncoated, rhodium-coated, and nitrified NiTi wires were immersed into artificial saliva of different pH values (4.8, 5.1, 5.5, and 6.6). Released nickel and titanium ions were measured with inductively coupled plasma-optical emission spectroscopy at the end of 28 days. Atomic force microscopy was used to measure the arithmetic average surface roughness *R*_a_, the root-mean-square roughness *R*_q_, and the maximum height of the asperities *R*_Z_. The nanoindentation hardness (*H*_IT_) and Young’s modulus (*E*_IT_) measurements were performed. The change in the pH of artificial saliva is inversely proportional to the release of titanium from both coated and uncoated wires, and the release of nickel from uncoated wires. The surface roughness parameters of both coated and uncoated wires are unaffected by the change in the pH of artificial saliva. The change in the pH of saliva has minor influence on the hardness and Young’s modulus of elasticity of both coated and uncoated wires. The concentration of released metal ions measured was below the recommended upper limit for daily intake; nevertheless, hypersensitivity effects cannot be excluded, even at lower concentrations and at low pH.

## 1. Introduction

The design of fixed orthodontic appliances can be complex, enabling plaque formation and accumulation [1], resulting in increases in the incidence and severity of the white spot lesions associated with orthodontic treatment [2]. Typical plaque retention sites are around the archwires, brackets (with archwires engaged in the bracket slots, connecting all brackets and tubes), and gingiva. The plaque pH remains low (4.8 or lower) for as long as the biofilm is undisturbed. Saliva flow and pH vary during the day, and, in patients with an increased number of *S. mutans* bacteria, periods of lowered pH of saliva happen more often because of the bacterial acid production [3]. Furthermore, in such patients, the periods of lowered pH are longer, i.e., in between meals and during sleep [4]. Therefore, most often, the pH of saliva fluctuates in the range from the low dental plaque to a near-neutral pH. The focus of this study will hence be on the changes caused by saliva in the pH range from 4.8 to 6.6.

The surface oxide coating on NiTi wires is mainly composed of titanium-dioxide (TiO_2_) which is stable within saliva of (near) neutral pH, preventing the release of allergen nickel into the oral cavity. In a lowered pH of saliva, the TiO_2_ shows signs of dissolution; still, the release of nickel remains under the allergenic threshold [5]. Titanium nitride (TiN) coatings on NiTi wires were introduced to increase the anticorrosive properties of common uncoated NiTi wires [6]. Improvement in the corrosion resistance was noted in short-term immersion tests, but in prolonged and repeated contact with fluoridated agents, the TiN coating dissolves and turns into TiO_2_, also displaying corrosive behavior [6,7,8]. It is not known how the change in the pH of saliva influences the protective character of the TiN coating. A whitish rhodium coating (a mixture of noble metals rhodium and gold) was introduced. In addition to protecting against the release of nickel, it works as a highly aesthetic solution [9]. The literature reports a quick loss of aesthetic appearance after oral exposure [8]. Furthermore, the nickel release in artificial saliva and low fluoride content prophylactic agents is increased, and could reach an allergenic threshold [5,7]. Previous research aimed to describe coatings on as-received wires in detail, and it was found that the coatings on both wire types were very thin [6,9]. It is not known, however, whether the change in the pH of saliva influences the dynamics of the release of ions from any type of coated NiTi wires. Furthermore, implications for human health, especially in relation to hypersensitivity and/or allergic thresholds, need to be determined.

Observation of the surface roughness properties of the orthodontic wires is important in determining the usage of certain kinds of wires in clinical work. Surface irregularities facilitate plaque accumulation, decrease the wire’s esthetic appearance, and promote the corrosion processes. Furthermore, surface texture and composition both determine the friction upon the brackets [1,6,8]. Atomic force microscopy (AFM) is a non-invasive method for quantitative and qualitative analysis of the surface roughness, which enables three-dimensional insight into the wires’ micromorphology. Previous research found that both uNiTi and RhNiTi had increased surface roughness parameters after clinical use. In fact, when compared to the uNiTi wires, the RhNiTi have increased surface roughness parameters even in as-received state [8,9].

On the other hand, mechanical properties, influencing the formability, the resilience, and the compliance are important in describing the working properties of the wires. If the corrosion processes are progressive, the changes in mechanical properties will be more pronounced [8]. For clinical work, it is important to know what to expect from certain types of wires, to be aware of the possible issues during exposure to various intraoral conditions [10], and to assess the significance of the findings for human health risks.

Therefore, the aim of this study was to explore whether changes in the salivary pH influence the mechanical properties, surface roughness, and ion release from the NiTi archwires with various surface coatings. The measured concentrations of specific metal ions released were compared to the recommended levels for daily intake.

## 2. Materials and Methods

Three types of superelastic nickel–titanium alloy orthodontic archwires with a 0.020-inch × 0.020-inch rectangular cross section were studied:-NiTi archwire with untreated surface (uNiTi), (Sentalloy, Dentsply GAC, Bohemia, NY, USA);-NiTi archwire marketed as rhodium (Rh)-coated (RhNiTi), (High Aesthetic, Dentsply GAC, Bohemia, NY, USA);-NiTi archwire with a titanium nitride (TiN) surface (NNiTi), (IonGuard, Dentsply GAC, Bohemia, NY, USA).

Specimens of each wire type were cut from the arch forms; the sample in each immersion tube was 100 mm long. Wire samples were immersed into an artificial saliva solution (1.5 g/L KCl, 1.5 g/L NaHCO_3_, 0.5 g/L NaH_2_PO_4_ × H_2_O, 0.5 g/L KSCN, 0.9 g/L lactic acid) for 28 days, while the testing tubes were incubated at 37 °C in a water bath. To simulate the daily variations in the saliva change, four solutions with different pH values of artificial saliva (4.8, 5.1, 5.5, and 6.6) were adjusted with lactic acid and NaOH [11]. Every experimental condition was triplicated.

The released nickel and titanium ions were measured by means of inductively coupled plasma optical emission spectroscopy (ICP-OES) using a PRODIGY Spectrometer (Teledyne Leeman Labs Inc., Hudson, NY, USA) at the end of the incubation period. The ions’ stability before measurement was achieved by adding one drop of ultra-pure HNO_3_. The detection limits for Ni and Ti were 2 and 5 ppb. Additional samples of salivary solutions without wire were made, which served as blank samples (negative controls). Data were expressed in µgcm^−2^. For human health risk assessment, the average amount of ions released if there were two (upper and lower) archwires exposed was calculated according to Arndt et al. [12], and expressed in µg.

Atomic force microscopy (AFM) was used to depict the surface morphology of the samples and to measure the arithmetic average surface roughness *R*a, the root mean square (RMS) roughness *R*q, and the maximum height of the asperities *R*_Z_. Contact-mode AFM measurements were performed by using a Dimension Icon SPM (Bruker, Karlsruhe, Germany). The measurements on the 30 µm × 30 µm surfaces were controlled via the instrument’s NanoScope software; this software was also used to flatten measurement data, filtering the inclination of the probe with respect to the surface of the sample. SNL-10 (Bruker, Karlsruhe, Germany) high-resolution probes with a 2 nm Si tip radius, mounted onto a 0.6 µm thick triangular Si_3_N_4_ cantilever (with a bending stiffness of 0.12 Nm^−1^), were used.

The nanoindentation hardness (*H*_IT_) and Young’s modulus (*E*_IT_) measurements were performed with the Keysight G200 Nanoindenter (force resolution: 50 nN, indentation resolution: 0.01 nm) according to the ISO 14577 standard; a 20 nm Berkovich tip was used. Measurements were conducted by indentation in a 4 × 4 pattern of points with a peak load of 20 mN (indentation depth of ~1 µm), followed by an indentation in a different area with a 100 mN peak force (indentation depth of ~2 µm). The loading time was 10 s, followed by a peak hold time of 1 s and allowable drift rates of 0.2 nms^−1^.

All experiments were conducted on three samples of each wire type/pH combination.

The ANOVA with Student–Newman–Keuls post hoc test, and Pearson correlations, were used for statistical analyses (SPSS 22, IBM, Armonk, NY, USA).

## 3. Results

### 3.1. Ion Release

#### 3.1.1. Nickel Ions Release

Table 1 presents data of the released Ni-ions from the used wires, after immersion in the saliva of pH ranging from 4.8 to 6.6. Two-way ANOVA showed that the wire type introduces significant variability in the Ni-ion release (*p* = 0.025; η^2^ = 0.263), while the pH, the combination of the wire type, and the salivary pH showed no importance. A significantly smaller Ni-ion release was noted in the pH 6.6 conditions for the uNiTi wire, compared to the other pH values (*p* = 0.018, η^2^ = 0.698). There was a significantly lower Ni-ion release at pH 5.5 for the NNiTi, when compared to the pH of 5.1 and 4.8 (*p* = 0.011, η^2^ = 0.735). Due to the dispersion of data, for the RhNiTi wire, no significant differences could be established. The RhNiTi wire released most of the Ni-ions at pH 4.8, 5.1, and 5.5, most notably at pH 5.5 (*p* = 0.001, η^2^ = 0.893); other data were scattered. When compared to uNiTi and RhNiTi, the NNiTi released significantly more Ni-ions at pH 6.6 (*p* = 0.017, η^2^ = 0.744).

#### 3.1.2. Titanium Ions Release

Table 2 presents data of the released Ti-ions from the used wires, after immersion in the saliva of pH ranging from 4.8 to 6.6. Two-way ANOVA showed that the cumulative Ti-ions’ release depends on the wire type (*p* < 0.001, η^2^ = 0.801), the pH (*p* < 0.001, η^2^ = 0.633), and the combination of the wire and the pH (*p* < 0.001, η^2^ = 0.692). The uNiTi released significantly more Ti-ions at pH 5.1 and 4.8, when compared to pH 5.5 and 6.6 (*p* < 0.001, η^2^ = 0.914). The RhNiTi wire released significantly more Ti-ions at pH 4.8, when compared to the other pH (*p* = 0.004, η^2^ = 0.798). At lower pH values (4.8, 5.1, and 5.5), the RhNiTi released significantly more Ti-ions, when compared to the uNiTi and NNiTi (*p* < 0.05, η^2^ = 0.727–931); at pH 6.6 the NNiTi released significantly more Ti-ions than the RhNiTi wire (*p* = 0.042, η^2^ = 0.653).

#### 3.1.3. Ions Release for Two Wires

Table 3 reports the data on the average release of Ni- and Ti-ions if there were two wires (for the upper and the lower jaw) in the mouth, with a total surface of 7.2 cm^2^ and 28 cm of length, according to the work of Arndt et al. [12].

### 3.2. Surface Roughness

Figure 1 depicts the surface roughness parameters for all three wire types after immersion in artificial saliva with four different pH values; the RhNiTi showed higher roughness values than the other two wires, but there were no significant differences between the values for the various saliva pH values. 

The rather large dispersion of data, related to optically visible material inhomogeneity, could be related to the production process of the wires; the typical AFM findings from every experimental group can be presented as a three-dimensional chart, as depicted in Figure 2.

### 3.3. Mechanical Properties

The obtained values of *H*_IT_ and *E*_IT_ (Figure 3) showed a certain amount of variety among the measurements for all the analyzed wires and pH conditions. Higher values were obtained using the larger force during testing (100 mN versus 20 mN), as the nanoindentation probe entered deeper into the tested material, past the surface irregularities. As in the case of surface roughness, a rather large dispersion of the nanoindentation values was observed.

### 3.4. Pearson Correlations

Pearson correlations were used to determine the relations between the observed parameters of metal ions’ release, surface roughness, hardness, and Young’s modulus of elasticity and the pH of the saliva (Table 4).

## 4. Discussion

### 4.1. The Human Health Risk Assessment—Hypersensitization to Nickel and Titanium

Human health risk assessment is the process of evaluating the potential impact of a hazard on a person’s health. In our study, the hazard is represented by the release of Ni-and Ti-ions and the individuals at risk are the patients receiving non-removable orthodontic appliances. The amounts of Ni- and Ti-ions for two wires were compared to the acceptable daily intake (ADI), which is the amount of a given substance in food or drinking water that can be ingested daily (orally) over a lifetime without posing a significant health risk [13]. The no observed adverse effect level (NOAEL), extrapolated from animal studies and observations in humans, is an experimentally determined dose at which there is no statistically or biologically significant evidence of the toxic effect, and is used to determine the ADI.

The NOAEL for Ni depends on the compound tested, and ranges from 10 µM for NiCl_2_ on human keratinocytes [14] to 100 µM for NiCl_2_-6H_2_O on BALB/3T3 cells [15] and 150 μM for Ni(NO_3_)_2_ on rat liver cells [16]. The oral reference dose of nickel would suggest a benchmark dose of 4–5 mg Ni/kg/day, based on increased prenatal mortality due to the oral ingestion of nickel sulfate and nickel chloride [17]. The NOAEL of the nickel ions released from nickel-containing medical devices is 0.25 mg/kg bw/day for SD rats, whereas the threshold of toxicological concern of nickel is 150 μg/day, based on the application of the 100-fold uncertainty factor and the body weight of a 60 kg person [18]. The RhNiTi in our research showed high amounts of released nickel. If we imagine those amounts added to full mouth appliance nickel release (from brackets, molar bands, and wire ligatures), the threshold of the NOAEL for Ni could thus be exceeded.

The NOAEL for Ti particles was, in turn, observed at 0.625 mg/mL on osteoblasts MC3T3-E1 [19]. None of the wires from our research reach the NOAEL for Ti by far.

The tolerable upper intake level (UL) for Ni, which is the highest daily nutrient intake unlikely to pose a risk of adverse health effects for almost all individuals in the general population, is set at 1.0 mg/day [20]. The derived no objection level (DNEL) is the level of exposure to a substance to which humans should not be exposed [21], and is set at 11 µg/kg bw/day for Ni [22] and 350 mg/kg bw/day for Ti [23]. Since there are many sources of nickel from the daily consumption of water, nuts, and grains, as well as from skin lotions, detergents, tattoos, piercing, jewelry, etc., it is advisable to keep the intake from additional known sources (such as orthodontic appliances, or dental prosthetic materials) as low as possible [24].

The accumulated nickel skin dose (μg/cm^2^) is estimated to be the most important factor determining the risk of nickel allergy and allergic nickel dermatitis [25,26]. Fischer et al. (2005) showed that 5% of a sensitized population react to 0.44 µg Ni/cm^2^ and 10% react to 1.04 µg Ni/cm^2^ [27]. Jensen et al. (2006) performed a meta-analysis to find the estimated thresholds of nickel doses that may cause systemic contact dermatitis in nickel-sensitive patients [28]. The results showed that the most sensitive groups may react with systemic contact dermatitis at normal daily nickel exposure from drinking water, or food of 0.22–0.35 mg nickel in 1% of those individuals. The most sensitive nickel allergic individuals reacted to stainless steel with an estimated nickel release equivalent to 0.01 μg/cm^2^ per week [29]. According to the EU Nickel Directive, nickel-containing products intended for direct and prolonged contact with the skin must not release more than 0.5 µg Ni/cm^2^/week [22]. Further implications for human health present overlying infections, as released Ni changes bacterial metabolism, and presents more complex clinical conditions [30]. Additionally, patients with sensitive skin, because of their defective skin barrier, have an increased absorption of nickel [30]. Titanium absorption in patients with different skin conditions or infections needs to be investigated in future research.

Although titanium is generally considered a non-allergenic material, some studies have found isolated cases of allergies in the vicinity of titanium-containing materials, e.g., dental implants, which may cause type IV (delayed) or I (immediate) reactions in allergic patients [31,32,33,34,35]. Titanium allergies are rare; in dental implant patients, a prevalence of 0.6% was detected [35]. The threshold limit value for hypersensitivity to Ti by oral or skin absorption could not be found. A titanium allergy could also result from the impurities in titanium, such as nickel, chromium, and cadmium. Due to scarce data on Ti-induced allergies, further studies are indicated. Our research marked 3.5–6.5× higher release of Ti per week from RhNiTi wires, when compared to conventional wires. Further observation of clinical manifestations is needed.

The release of titanium is inversely proportional to the change in the pH of saliva for all three tested wire types (uNiTi, RhNiTi, and NNiTi). The release of nickel is inversely proportional to the change in the pH of saliva only for the uNiTi wires. The NNiTi released overall low amounts of nickel, similar to those of the uNiTi, while the RhNiTi showed great variability among specimens. This was already indicated in a clinical study, which reported on a wide variety of morphological changes observed in clinical use [36]. The higher release of nickel from the RhNiTi was observed during another experimental setting and recorded on another instrument, confirming suspicions on uneven and non-homogeneous coatings, which promote corrosion in the defects within noble coatings [37]. The chemical compositions of the wires’ surface, accompanied by the non-homogenous nature of the coating, plus fresh cuts at the wire’s ends (which present additional areas at which promoted corrosion occurs, but are an inevitable part of daily work in adjusting the wire to an individual’s needs), are all responsible for the accelerated corrosion on the rhodium- and gold-coated wires. These findings indicate the possibility of a wide variety of responses in clinical use, and call for precautions in persons with a known sensibility to nickel.

The measured concentrations of specific metal ions released from orthodontic wires were not above the recommended levels for daily intake. Nevertheless, hypersensitivity reactions and possible synergistic effects of metal mixtures should be considered. In addition, a higher daily dose of metal ions could be obtained if exposure to other sources (e.g., food intake, cosmetics, etc.) is considered.

### 4.2. Surface and Mechanical Properties

The change in the pH of saliva does not significantly change the surface roughness parameters of all the three considered wire types. Surface roughness parameters are highest for the RhNiTi, from all three observed wires. In fact, the RhNiTi wires were the roughest in the as-received condition, and they maintained the roughest surface after immersion in artificial saliva of various pH values, as well as after clinical use [8,9,37]. It is important to note that surface roughness is associated with an increased susceptibility to corrosion, which results in an increased release of metals. As indicated above, the increased amounts of nickel and titanium into the oral cavity and adjacent soft tissues [38,39] could pose a danger in direct contact with cells, either in the development of contact allergies, or by modifying the immune response related to atopic allergy via Staphylococcal infections [30,40].

The change in the pH of the saliva does not induce a significant change in the mechanical properties of the uNiTi, while it does affect the coated NiTi wires. It was observed that the results obtained with the nanoindentation technique are more sensitive to surface roughness [41], which was increased in both coated wires; therefore, the observations obtained in this work could be attributed to the coating, and not the bulk material. The composition of the surface layer also contributes to the results recorded with nanoindentation techniques [42]. The Pearson correlations also indicate that the change in salivary pH could account for around 30% of the variability in the mechanical properties of the coated wires. Therefore, one should take precautions in translating those data into clinical recommendations.

The limitations of this study are in its observation of the changes caused by saliva with various pH; additional information about changes of the surfaces and electrochemical processes could be obtained with X-ray diffraction (XRD), scanning electron microscopy (SEM), and electrochemical testing in the future.

## 5. Conclusions

The change in the pH of artificial saliva is inversely proportional to the release of titanium from both coated and uncoated wires, and the release of nickel from uncoated wires.

The surface roughness parameters of both coated and uncoated wires are unaffected by the change in the pH of artificial saliva.

The change in the pH of saliva has minor influence on the hardness and Young’s modulus of elasticity of both coated and uncoated wires.

Lower release of both nickel and titanium is expected in saliva close to neutral pH. If oral hygiene is compromised, coated RhNiTi wires are to be avoided in order to minimize the risk of adverse effects on human health and hypersensitivity-related complications.

## Figures and Tables

**Figure 1 materials-15-01994-f001:**
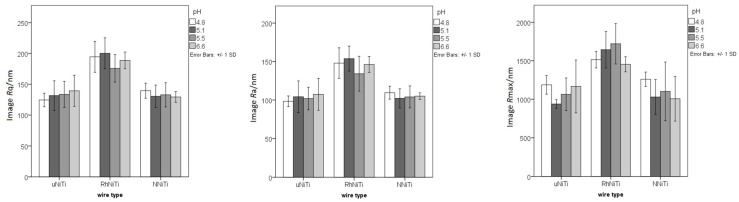
The surface roughness parameters (axis y: *R*q/nm, *R*a/nm, *R*max/nm) for the uncoated NiTi (uNiTi), rhodium-coated NiTi (RhNiTi), and nitrified NiTi (NNiTi) wires after immersion in artificial saliva with four different pH values.

**Figure 2 materials-15-01994-f002:**
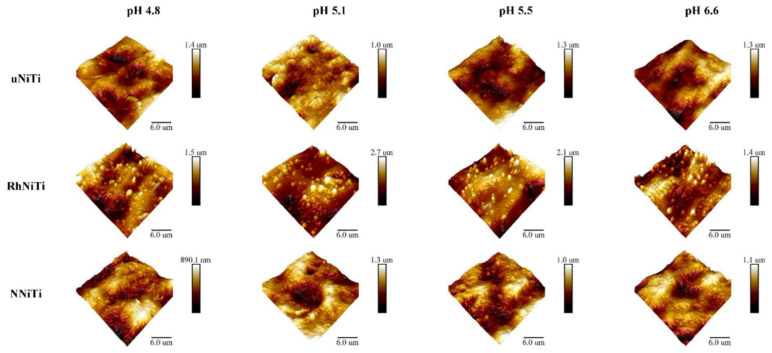
Typical AFM findings from every experimental group (horizontal rows: uncoated NiTi (uNiTi), rhodium-coated NiTi (RhNiTi), and nitrified NiTi (NNiTi) wires), after immersion in artificial saliva with four different pH values (vertical columns), presented as a three-dimensional chart.

**Figure 3 materials-15-01994-f003:**
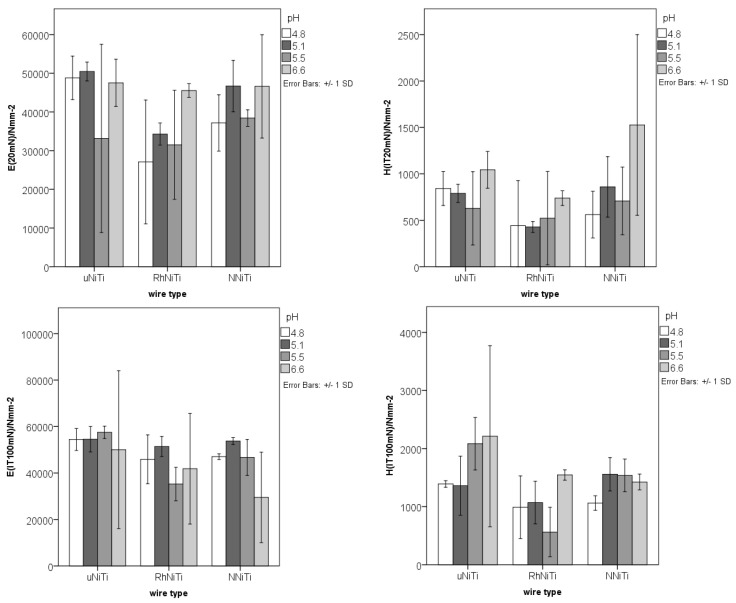
The nanoindentation hardness (*H*_IT_) and Young’s modulus (*E*_IT_) values (shown on “y” axes) for the uncoated NiTi (uNiTi), rhodium-coated NiTi (RhNiTi), and nitrified NiTi (NNiTi) wires, after immersion in artificial saliva with four different pH values.

**Table 1 materials-15-01994-t001:** Distribution of the cumulative Ni-ion release in relation to the pH of artificial saliva and the wire type.

Wire Type	pH	AM (SD) (µgcm^−2^)	*p*	η^2^
uNiTi	4.8	0.55 (0.10) ^a^	0.018	0.698
	5.1	0.49 (0.02) ^a^		
	5.5	0.59 (0.15) ^a^		
	6.6	0.29 (0.06) ^b^		
NNiTi	4.8	1.45 (0.35) ^a^	0.011	0.735
	5.1	1.20 (0.14) ^a^		
	5.5	0.50 (0.03) ^b^		
	6.6	0.86 (0.37) ^a,b^		
RhNiTi	4.8	144.83 (153.94)	0.567	0.107
	5.1	8.73 (5.96)		
	5.5	34.03 (11.59)		
	6.6	0.18 (0.02)		

uNiTi, uncoated NiTi; RhNiTi, rhodium-coated NiTi; NNiTi, nitrified NiTi; AM (SD), arithmetic mean (standard deviation); *p*, statistical significance; η^2^, power of effect, ^a, b^ different superscript letters denote statistically significant differences determined by the ANOVA and Student–Newman–Keuls post hoc test.

**Table 2 materials-15-01994-t002:** Distribution of the cumulative Ti-ions release in relation to the pH of artificial saliva and the wire type.

Wire Type	pH	AM (SD) (µgcm^−2^)	*p*	η^2^
uNiTi	4.8	0.42 (0.11) ^a^	<0.001	0.914
	5.1	0.61 (0.10) ^b^		
	5.5	0.17 (0.04) ^c^		
	6.6	0.07 (0.04) ^c^		
NNiTi	4.8	0.34 (0.13)	0.036	0.636
	5.1	0.40 (0.21)		
	5.5	0.11 (0.00)		
	6.6	0.11 (0.01)		
RhNiTi	4.8	4.20 (1.42) ^a^	0.004	0.798
	5.1	2.16 (1.00) ^b^		
	5.5	2.04 (0.51) ^b^		
	6.6	0.05 (0.02) ^c^		

uNiTi, uncoated NiTi; RhNiTi, rhodium-coated NiTi; NNiTi, nitrified NiTi; AM (SD), arithmetic mean (standard deviation); *p*, statistical significance; η^2^, power of effect, ^a, b, c^ different superscript letters denote statistically significant differences determined by the ANOVA and Student–Newman–Keuls post hoc test.

**Table 3 materials-15-01994-t003:** Average release of nickel (Ni) and titanium (Ti) ions for a full mouth after one and four weeks of immersion in artificial saliva with pH of 4.8, 5.1, 5.5, and 6.6.

Ions (µg)	Wire Type	pH	2 Wires/1 Week	2 Wires/4 Weeks
Ni	uNiTi	4.8	0.99	3.96
		5.1	0.88	3.53
		5.5	1.06	4.25
		6.6	0.52	2.09
	NNiTi	4.8	2.61	10.44
		5.1	2.16	8.64
		5.5	0.90	3.60
		6.6	1.55	6.19
	RhNiTi	4.8	260.69	1042.78
		5.1	15.71	62.86
		5.5	61.25	245.02
		6.6	0.32	1.30
Ti	uNiTi	4.8	0.76	3.02
		5.1	1.10	4.39
		5.5	0.31	1.22
		6.6	0.13	0.50
	NNiTi	4.8	0.61	2.45
		5.1	0.72	2.88
		5.5	0.20	0.79
		6.6	0.20	0.79
	RhNiTi	4.8	7.56	30.24
		5.1	3.89	15.55
		5.5	3.67	14.69
		6.6	0.09	0.36

uNiTi, uncoated NiTi; NNiTi, nitrified NiTi; RhNiTi, rhodium-coated NiTi.

**Table 4 materials-15-01994-t004:** Pearson correlations between the released amount of nickel (Ni) and titanium (Ti) ions; the arithmetic average surface roughness *R_a_*; the RMS roughness *R_q_*; the maximum height of the asperities *R_Z_*; the nanoindentation hardness (*H_IT_*) at 20 mN (*IT*20 mN) and 100 mN (*IT*100 mN); and Young’s modulus (*E_IT_*) at 20 mN (*IT*20 mN) and 100 mN (*IT*100 mN); and the pH of saliva, for the uncoated (uNiTi), rhodium-coated (RhNiTi), and the nitride-coated (NNiTi) nickel–titanium orthodontic archwires.

	uNiTi		RhNiTi		NNiTi	
	pH	*p*	pH	*p*	pH	*p*
Ni/µgcm^−2^	**−0.688 ***	**0.013**	−0.448	0.144	−0.485	0.11
Ti/µgcm^−2^	**−0.765 ***	**0.004**	**−0.837 ***	**0.001**	**−0.614 ***	**0.034**
*R_q_*/nm	0.276	0.386	−0.158	0.624	−0.213	0.507
*R_a_*/nm	0.198	0.538	−0.098	0.762	−0.082	0.8
*R_z_*/nm	0.142	0.659	−0.229	0.473	−0.284	0.372
*E_IT_* (*IT*20 mN)/Nmm^−2^	−0.061	0.852	**0.573 ***	**0.051**	0.295	0.351
*H_IT_* (*IT*20 mN)/Nmm^−2^	0.357	0.254	0.388	0.213	**0.589 ***	**0.044**
*E_IT_* (*IT*100 mN)/Nmm^−2^	−0.124	0.701	−0.193	0.547	**−** **0.649 ***	**0.022**
*H_IT_* (*IT*100 mN)/Nmm^−2^	0.422	0.172	0.449	0.144	0.272	0.392

* statistically significant findings.

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
