# Peer review of "Ion Release and Surface Changes of Nickel–Titanium Archwires Induced by Changes in the pH Value of the Saliva—Significance for Human Health Risk Assessment"

_materials, 2022, doi:10.3390/ma15061994_

Round 1

Reviewer 1 Report

In my opinion the manuscript (materials-1604688) entitled ‘Ion release and surface changes of nickel-titanium archwires induced by changes of the pH value of the saliva – significance for human health risk assessment’ can be recommended for publication in Materials after minor revision:

  • After reading the publication, I have doubts about the scientific novelty of the manuscript. Therefore, the Authors should clearly indicate what is the scientific novelty of their paper.
  • Did the authors also test the surface roughness and mechanical properties of blanks - wires that were not exposed to saliva?
  • 315 - "[4]" should be replaced by " [41]"  

Author Response

Thank the reviewers for their comments.

Reviewer 1

In my opinion the manuscript (materials-1604688) entitled ‘Ion release and surface changes of nickel-titanium archwires induced by changes of the pH value of the saliva – significance for human health risk assessment’ can be recommended for publication in Materials after minor revision:

  • After reading the publication, I have doubts about the scientific novelty of the manuscript. Therefore, the Authors should clearly indicate what is the scientific novelty of their paper.

Response: Novelty of the paper is the observation of changes in the NiTi wires with various coating caused by saliva with different pH values, as it has not been described yet. It is stated in the Introduction section. Additional text is also in the Conclusion section, to emphasise findings from this research.

  • Did the authors also test the surface roughness and mechanical properties of blanks - wires that were not exposed to saliva?

Response: Blanks (as-received wires) were not tested in this experiment, focus was on changes caused by different pH of saliva, and comparison between those groups. There is text added regarding previous research on blanks.

  • 315 - "[4]" should be replaced by " [41]"  

Response: The reference number is corrected.

Reviewer 2 Report

Osmani et al present an interesting and practical study on the performance of nickel-titanium wire used for dental devices after soaking in artificial saliva. The work is intended to shed light on the safety of commercial TiNi wire modified with various coatings. This idea and the resulting information deserves publication. However, for the Materials Journal, a more thorough description of the studied materials and an explanation of the results will be required.

The Major comments includes:

1) It is necessary to provide the chemical composition of the studied coatings. Probably, the materials correspond to the previous work (ref. [6]). Then it is incorrect to call one of the types of coating "rhodium-coated" (line 84), because this is a rhodium alloy coating, not pure rhodium.

2) The morphology of the original surface of bare TiNi wires and the microstructure and thickness of both types of coated wires should be characterized in details. In particular, the structure of the rhodium-contained coating (surface and cross-section) may be the reason for the observed results in the release of metal cations. It is desirable to use scanning electron microscopy (SEM) and X-ray diffraction (XRD).

3) It is not clear how to refer to the data in Table 3 in comparison with the data in Tables 1 and 2. In particular, the first two tables have a different dimension and contain a standard deviation. Please modify Table 3 in the same way.

4) It is not clear what data are shown in the individual graphs in Figures 1 and 3. All three charts in Figure 1 and all 4 charts in Figure 3 are different. And they all contain data for all four pH and all three types of wires. Please make it clear which factor changed in each chart. Probably, these are statistical data for different studied samples…

5) It is necessary to offer a clearer reason for the results obtained than "chemical composition". Why does the composition of the coating have such an effect. First of all, I mean rhodium-contained coating, which was found to be unacceptable for use. For example, it was recently observed [Vikulova, et al. "MOCVD of Noble Metal Film Materials for Medical Implants: Microstructure and Biocompatibility of Ir and Au/Ir Coatings on TiNi." Coatings 11.6 (2021): 638] that with a certain microstructure of the iridium coating, electrochemical activation of nickel precipitation occurs. Rhodium is the electronic analog of iridium, so could something similar be the cause in your work? Provide evidence to support/refute this idea (see point 2).

6) In continuation of this idea, it is necessary to explain whether the pieces (10 mm and 28 cm) were cut off from the wire. If the ends of the wire were exposed due to cutting, then a fresh electrochemical contact was obtained (rhodium cathode and nickel/titanium soluble anode). Then the results of the experiments do not correspond to what will be observed with a completely covered wire.

7) The conclusion that it is better not to use all types of coatings is not clear (lines 321-322). The titanium nitride coating seemed to perform even better (Table 1).

Technical points:

1) It is necessary to make more visible signatures/legends on all Figures. This is especially true for AFM images and roughness data.

2) In the introduction, it is better to first briefly present where the wire is used, and then describe its surface composition (line 42).

3) Provide information for sample preparation and calibration solutions for ICP-OES.

Author Response

Thank the reviewers for their comments.

Reviewer 2

Osmani et al present an interesting and practical study on the performance of nickel-titanium wire used for dental devices after soaking in artificial saliva. The work is intended to shed light on the safety of commercial TiNi wire modified with various coatings. This idea and the resulting information deserves publication. However, for the Materials Journal, a more thorough description of the studied materials and an explanation of the results will be required.

The Major comments includes:

1) It is necessary to provide the chemical composition of the studied coatings. Probably, the materials correspond to the previous work (ref. [6]). Then it is incorrect to call one of the types of coating "rhodium-coated" (line 84), because this is a rhodium alloy coating, not pure rhodium.

Response: Text is added int the introduction and Materials and Methods section. Chemical composition of coatings was described previously by, Iijima et al. (2012), and Iijima et al. (2010). The reviewer is right, there are other metals in the High aesthetic wires, and it is noted in the text in the Introduction section. The specimens were named “rhodium-coated”, because the high aesthetic wire was marketed as Rh-coated, and whitish appearance was attributed to rhodium.

2) The morphology of the original surface of bare TiNi wires and the microstructure and thickness of both types of coated wires should be characterized in details. In particular, the structure of the rhodium-contained coating (surface and cross-section) may be the reason for the observed results in the release of metal cations. It is desirable to use scanning electron microscopy (SEM) and X-ray diffraction (XRD).

Response: Previous research characterised the mentioned coatings in details. Therefore, this research focused on changes caused by immersion in saliva with various pH values. We agree that SEM and XRD would provide additional information, and it is stated so under the limitations of this study.

3) It is not clear how to refer to the data in Table 3 in comparison with the data in Tables 1 and 2. In particular, the first two tables have a different dimension and contain a standard deviation. Please modify Table 3 in the same way.

Response: Data in Table 2 show results from the ICP-OES measurements, and are expressed as μgcm-2, so that they can be used to calculate expected amount of ions released from different dimensions of the same type of wire (either one of the three types used in this study). Data in Table 3 are calculated to depict average clinical condition (if two wires, for upper and lower dental arch, were mounted, in total 28 cm long). Data are expressed as μg of ions, as this is average value expected, used for the assessment of health hazards.

4) It is not clear what data are shown in the individual graphs in Figures 1 and 3. All three charts in Figure 1 and all 4 charts in Figure 3 are different. And they all contain data for all four pH and all three types of wires. Please make it clear which factor changed in each chart. Probably, these are statistical data for different studied samples…

Response: In charts are presented data for different variables for all three wire types and all four pH values. Variables are inscribed on axis y on every chart. Additional explanations are added in Figure legends.

5) It is necessary to offer a clearer reason for the results obtained than "chemical composition". Why does the composition of the coating have such an effect. First of all, I mean rhodium-contained coating, which was found to be unacceptable for use. For example, it was recently observed [Vikulova, et al. "MOCVD of Noble Metal Film Materials for Medical Implants: Microstructure and Biocompatibility of Ir and Au/Ir Coatings on TiNi." Coatings 11.6 (2021): 638] that with a certain microstructure of the iridium coating, electrochemical activation of nickel precipitation occurs. Rhodium is the electronic analog of iridium, so could something similar be the cause in your work? Provide evidence to support/refute this idea (see point 2).

Response: Previous research explained why Rh/Au coating on NiTi wire released more ions in saliva with neutral pH, when compared to bare NiTi wires (37), and this was also stated in the Discussion section and in the limitations of this study.

6) In continuation of this idea, it is necessary to explain whether the pieces (10 mm and 28 cm) were cut off from the wire. If the ends of the wire were exposed due to cutting, then a fresh electrochemical contact was obtained (rhodium cathode and nickel/titanium soluble anode). Then the results of the experiments do not correspond to what will be observed with a completely covered wire.

Response: The wires used in experiment were cut into 10cm long pieces, fresh cuts are exactly the same as in clinical conditions and its surface was calculated into formulas when expressing released ions as μgcm-2. Data in Table 3 are calculated to depict clinical condition (if two wires, for upper and lower dental arch, were mounted, in total 28 cm long). In clinical use wire’s ends are always exposed with fresh cuts, when adjusting to fit individual arch length. The scope of this trial was to imitate intraoral conditions, and note what can be expected in clinical situation (in terms of ions released). Additional text considering this issue is in the Materials and Methods and the Discussion sections.

7) The conclusion that it is better not to use all types of coatings is not clear (lines 321-322). The titanium nitride coating seemed to perform even better (Table 1).

Response: The Conclusion section has lines added, to emphasise main findings from this research.

Technical points:

1) It is necessary to make more visible signatures/legends on all Figures. This is especially true for AFM images and roughness data.

Response: New Figure was made for AFM, more clear legends are made for all Figures.

2) In the introduction, it is better to first briefly present where the wire is used, and then describe its surface composition (line 42).

Response: The Introduction was changed accordingly your suggestions.

3) Provide information for sample preparation and calibration solutions for ICP-OES.

Response: Ions stability before measurement was achieved by adding one drop of ultra-pure HNO3. The detection limit for Ni and Ti was 2 and 5 ppb. Additional samples of salivary solutions without wire were made, which served as a blank samples (negative controls). This was added to the Materials and Methods section.